# Serine/Threonine Protein Kinase STK16

**DOI:** 10.3390/ijms20071760

**Published:** 2019-04-10

**Authors:** Junjun Wang, Xinmiao Ji, Juanjuan Liu, Xin Zhang

**Affiliations:** 1High Magnetic Field Laboratory, Key Laboratory of High Magnetic Field and Ion Beam Physical Biology, Hefei Institutes of Physical Science, Chinese Academy of Sciences, Hefei 230031, China; wjunjun@mail.ustc.edu.cn (J.W.); xinmiaoji@hmfl.ac.cn (X.J.); 2Science Island Branch of Graduate School, University of Science and Technology of China, Hefei 230026, China; 3School of Life Sciences, Anhui University, Hefei 230601, China; 4Institutes of Physical Science and Information Technology, Anhui University, Hefei 230601, China

**Keywords:** STK16, kinase, fatty acylation, phosphorylation, Golgi apparatus, protein secretion and sorting, cell cycle

## Abstract

STK16 (Ser/Thr kinase 16, also known as Krct/PKL12/MPSK1/TSF-1) is a myristoylated and palmitoylated Ser/Thr protein kinase that is ubiquitously expressed and conserved among all eukaryotes. STK16 is distantly related to the other kinases and belongs to the NAK kinase family that has an atypical activation loop architecture. As a membrane-associated protein that is primarily localized to the Golgi, STK16 has been shown to participate in the TGF-β signaling pathway, TGN protein secretion and sorting, as well as cell cycle and Golgi assembly regulation. This review aims to provide a comprehensive summary of the progress made in recent research about STK16, ranging from its distribution, molecular characterization, post-translational modification (fatty acylation and phosphorylation), interactors (GlcNAcK/DRG1/MAL2/Actin/WDR1), and related functions. As a relatively underexplored kinase, more studies are encouraged to unravel its regulation mechanisms and cellular functions.

## 1. Introduction

The human protein kinases belong to a large family that consists of more than 500 members [1,2]. They are involved in the regulation of many different cellular processes, such as cell growth, survival, proliferation, apoptosis, and metabolism [3,4,5,6]. These kinases have homologous catalytic domains that consist of 250–300 amino acid residues [7,8]. Polymerase chain reactions using primers derived from the conserved sequences were used to amplify new kinase families [9,10]. In this way, Ligos et al. were the first to identify STK16 (Ser/Thr kinase 16), a member of a new subfamily of Ser/Thr kinases, which was named PKL12 (protein kinase expressed in day 12 fetal liver), from mouse fetal liver (E12) [11]. It was also known as Krct (kinase related to cerevisiae and thaliana) [12], EDPK (embryo-derived protein kinase) [13], MPSK (myristoylated and palmitoylated Ser/Thr protein kinase) [14] and TSF-1 (TGF-β stimulated factor 1) [15] due to its diverse discovery experiences, but conserved among vertebrates. A series of studies on the characterization and biological function analysis of this kinase have gradually revealed that it is actually a novel member of the NAK (Numb-associated family of protein kinases) kinase family with unique features and functions [16,17]. Although twenty years have passed since the discovery of STK16, it is still a relatively underexplored kinase. However, recent studies have revealed its important roles in cell signaling, cell cycle, Golgi assembly regulation, and protein secretion and sorting. This review aims to provide a comprehensive review about the STK16 molecular structure, expression, post-translational modification, interactors/substrates, and its cellular functions.

## 2. Distribution and Characterization of STK16

### 2.1. Distribution

STK16 is widely distributed and its homologues exist in eukaryotes from yeasts to humans, although most studies focused on the human and mice homologues. Moreover, *stk16* was identified on chromosome 1 in mice from Mouse Genome Informatics (http://www.informatics.jax.org) but localized to human chromosome 2 according to the Gene Ontology Annotation (GOA) database (https://www.ebi.ac.uk/GOA). Based on in situ hybridization, Northern blot, and RT-PCR, it was found that STK16 was widely expressed throughout murine development and in adult tissues on the mRNA level [11,12,13,14,15]. In addition, although having a ubiquitous distribution, STK16 is expressed preferentially and specifically within multiple tissues, in particular within the liver, kidney, and testis [11,12,13,15]. Furthermore, STK16 expression within the same tissue is higher in certain cell types. For example, compared with the mesenchymal compartments, STK16 expression level was higher in epithelial compartments [12,18]. Western blot results showed that STK16 was expressed in various cell lines, but it preferred to be expressed in adherent cells compared with suspension cells [19]. In cells, STK16 mainly localizes to the Golgi [11,19,20,21] and also enters the nucleus under certain circumstances [19].

### 2.2. Molecular Characterization of STK16 

STK16 homologues in various species share different similarity. Amino acid alignment results showed that the sequence similarity was greater than 90% in different vertebrates but only 29% to the *Saccharomyces cerevisiae* STK16 homologue named ENV7 [22]. The ORF (open reading frame) of mammalian STK16 contains 915 nucleotides and encodes 305 amino acids. It comprises a complete kinase catalytic domain, a very short N-terminal domain, and a brief C-terminus. Alignment of the kinase domain of STK16 with others illustrated that it is a new Ser/Thr kinase, displaying a conservative sequence element of Ser/Thr kinase, but lacking the necessary amino acids in subdomain VIb and VIII that are conserved in tyrosine kinases [11,12,13,14]. Activation segment sequences analysis together with secondary structure prediction suggested that it is distantly related to the other kinases and belongs to the family of human NAK [16]. According to the classification and phylogenetic analysis of Human Kinome, the NAK family does not belong to any group [2,23]. In fact, the crystal structure of STK16 was solved by Eswaran et al. in 2008 revealing the atypical activated loop ASCH (activation segment C terminal helix) in its catalytic domain, which also proved this classification [17].

Besides STK16, the human NAK family also includes AAK1 (adaptor-associated kinase 1), BIKE/BMP2K (BMP-2-inducible kinase), and GAK (cyclin G-associated kinase). Though their catalytic domains sequences have less than 30% similarity and share almost no conservation outside the kinase domain [16,17], their crystal structures solved by the Stefan Knapp group showed an ASCH architecture in all human NAKs (Figure 1) [17]. It provides new ideas and insights for a comprehensive understanding of the molecular characteristics of NAK family proteins and their related physiological functions. The assembly of the regulation region independent of atypical activation loop phosphorylation explains why NAKs are constitutively active. Moreover, the variable substrate binding grooves of NAKs suggest that they could participate in a broad range of cellular functions through interacting with different substrates. 

Among the kinases in the NAK family, STK16 is the less studied one. However, since the NAK kinases share the same distinctive ASCH structure, studies about the other family members can help us to understand the function of STK16. For example, in Drosophila, NAK interacts with the phosphotyrosine binding domain (PTB) of Numb, regulates asymmetric cell division and confers distinct fates to daughter cells [24,25,26]. In humans, AAK1 binds directly to adaptin, the membrane-tethered active form of Notch, or Numb, to regulate a variety of activities including clathrin-mediated endocytosis [27,28], Notch signaling pathway [29], or coated pit maturation [30]. Another member BIKE/BMP2K was found to play an important regulatory role in endocytosis associated with Numb [31]. Similarly, GAK, interacting with cyclin G-CDK5, is a potential regulator of clathrin-mediated membrane trafficking and mediates binding of clathrin and adaptors to the plasma membrane and the trans-Golgi network [32,33]. NAKs have been implicated in many diseases and supposed to be potential drug targets. For example, inhibition of AAK1 kinase has become a novel therapeutic approach to treat neuropathic pain, schizophrenia, Parkinson’s disease, and hepatitis C virus infection [34,35,36]. GAK has been discussed as a potential drug target for the treatment of viral infections due to its involvement in entry and production of multiple viruses [36,37,38]. Furthermore, BIKE was reported as a cellular factor associated with HIV-1 replication [39]. Recently, STK16 was also found to participate in cell division, cell signaling, and protein secretion and sorting, which will be discussed in detail later. STK16 IN-1, a small molecule inhibitor with high selectivity, has been developed to specifically inhibit the kinase activity of STK16 [40]. STK16-IN-1 could reduce cancer cell numbers and potentiate the antiproliferative effects of some chemotherapeutics [40]. Thus, inhibition of STK16 is expected to be developed as a novel therapeutic approach for cancers.

## 3. Posttranslational Modification of STK16 

### 3.1. STK16 Undergoes Specific Fatty Acylation Modification

Many protein kinases can be modified by myristic acid or palmitic acid to regulate fundamental biological processes of mammalian cells [41,42,43,44]. Myristoylation occurs in the cytosol, where the 14-carbon saturated fatty acid myristate is linked to an N-terminal Gly of the protein by a stable amide bond. Cys-palmitoylation, also known as *S*-palmitoylation, the 16-carbon saturated fatty acid palmitate is covalently linked to one or more specific Cys residues on the side chain via unstable thioester bonds [45,46]. Although both modifications could usually guide proteins to bind membrane, only the palmitoylation is reversible, which indicates that it could dynamically regulate kinase functions [47]. The best example is that Ras proteins are directed to the correct intracellular organelles for trafficking and perform activity just by palmitoylation [48]. In addition, it is reported that palmitoylation of three cysteines at the C-terminus of GRK6 directs its membrane binding and further regulates its subcellular distribution [49]. In vivo, both fatty acylations are usually coordinated and jointly regulate protein localization and function. Myristoylation first occurs at the N- terminal amino acids and this enables the protein to approach the membrane-bound palmitoyl acyltransferases and gets palmitoylated. Src family kinases [50,51] and AKAPs (A-kinase anchoring proteins) have been identified to be acylated by this way [52]. Another example is CDPK, a rice calcium-dependent protein kinase, whose myristoylation is essential for membrane localization, and palmitoylation is requisite for its full association [53].

STK16 may regulate its subcellular localization through fatty acylation modification. STK16 have a Gly at the position 2 and Cys at the position 6 and 8 of the N-terminus, respectively (Figure 2). These three conserved amino acids could be acylated by myristic acid and palmitic acid. In G2A (Gly 2 to Ala) mutation, myristoylation and palmitoylation were abolished. However, in C6,8S (Cys 6 and 8 to Ser) mutation, myristoylation was still detected but palmitoylation was completely lost. Therefore, it is suggested that the myristoylation of Gly2 in STK16 is necessary for the palmitoylation of Cys6 and Cys8 in this protein [14]. The homologue ENV7 in S.cerevisiae also has conservative palmitoylation sites at the N-terminus. Mutations at these sites result in palmitoylation abolishment and altered subcellular localization [22,54]. Besides, further analysis revealed that these acylation sites of ENV7 were not redundant and function in regulating ENV7′s stability, localization, phosphorylation, and vacuolar fusion, respectively [55]. Although this evidence indicates that ENV7 may perform different physiology functions through fatty acylation, the fatty acylation regulation of mammal STK16 homologue is still unknown.

### 3.2. STK16 is a Constitutively Active Ser/Thr Protein Kinase

In vitro, STK16 is capable of both autophosphorylation and phosphorylation of other substrates, such as MBP (myelin basic protein), Histone H1, PHAS-1, enolase, and Elk [11,12,13,14,15]. Phospho-amino acid analysis showed that the autophosphorylation sites of STK16 were mainly on Thr [14], whereas substrates of STK16 are phosphorylated mainly at Thr and Ser [11,13,14]. Screening a peptide library, an optimal substrate sequence of X-X-P/V/I-Φ-H/Y-T*-N/G-X-X-X (where Φ is an aliphatic residue, and * stands for the phosphorylated residue) was determined for STK16 [17]. This is consistent with the results of crystal analysis of STK16 [17]. Moreover, tagged or untagged STK16 protein, such as GST-, His-, and V5-tag, whether expressed and purified by the prokaryotic system (*Escherichia coli*) or the eukaryotic system (yeast, Sf9, Cos-1, or IP (immunoprecipitation) products from COS-7 cells transfected with STK16 cDNA), are all active kinase. The only two kinase-dead mutants generated in previous studies are K49M (Lys49 is in the ATP-binding site of STK16) and E202A (Glu202 is a key site to maintain the activation segment of kinase) [15,21,56].

In many protein kinases, phosphorylation on the Ser, Thr, or Tyr residues in the activation loop is required. The negatively charged phosphate groups compensate for the high positive charge in the activation segment, and the catalytic loop HRD (His-Arg-Asp residues in the catalytic loop) motif maintains the stability of the activation segment through polar interaction. Thus, activation loop phosphorylation is the most common method for regulating kinase activity [56,57]. Interestingly, the HRD arginine (Arg147) in STK16 forms a large number of polar interactions with the residues of the activation segment. In addition, a cluster of hydrophobic residues are close and stabilize the activation loop. Therefore, STK16 has a well-ordered catalytic conformation of the activation loop without dependence on phosphorylation or exogenous induction and is constitutively active (Figure 1) [17]. In fact, many kinases become constitutively active to be involved in the specific signaling pathway. For example, Akt becomes constitutively active when directed to the membrane by myristoylation, and this change induces glucose uptake into adipocytes in the absence of insulin and directs lipid synthesis [58]. Moreover, NIK (NF-κB-inducing kinase) is usually present as an autoinhibited form, and its constitutively active kinase domain is blocked by the inhibitory element [59]. After cytokine induction, NIK undergoes conformational changes and regulates NF-κB signal transduction through constitutive kinase activity [60]. However, the hyperactive NIK leads to autoimmune disease and cancer [61].

STK16 is capable of autophosphorylation at Thr185, Ser197, and Tyr198 of the activation segment (Figure 2). In the structure of unphosphorylated STK16, Ser197 and Tyr198 are buried in the hydrophobic cleft, resulting in slow phosphorylation of these two sites [17]. Besides these sites, there are also other potential phosphorylation sites in the activation loop of STK16, including Ser169, Ser180, and Thr95, which has not been investigated yet. Since Ser/Thr kinases usually possess multiple phosphorylation residues on the activation loop that have distinct effects on autophosphorylation [62,63,64], whether these sites are involved in the STK16 autophosphorylation and kinase activity needs further investigations.

### 3.3. The Relationship between Fatty Acylation and Phosphorylation of STK16

Many protein kinases are cooperatively regulated by fatty acylation and phosphorylation. For example, palmitoylated GRK6 has significantly increased kinase activity more than nonpalmitoylated wild-type GRK6 and a nonpalmitoylatable mutant GRK6 [65]. The viral protein TLCYnV C4 (Tomato leaf curl Yunnan virus C4) shuttles between the nucleus and the cytoplasm. Phosphorylated TLCYnV C4 promotes its myristoylation, then C4 achieve nuclear export [66]. For MARCKS (myristoylated alanine-rich protein kinase C substrate), myristoylation is the basis for its membrane localization. However, its phosphorylation by PKC promotes its rapid dissociation [67,68]. The mutations of all three cysteines (Cys13, 14, 15) or two of them at the N-terminal Env7, the yeast homologue of STK16, not only cause changes in membrane localization, but also affect its kinase activity by significantly decreasing the autophosphorylation level [54,55]. All this evidence suggests that there is some interplay between the phosphorylation and fatty acylation of these kinases. The relationship between the phosphorylation and fatty acylation of mammalian STK16 is currently unknown.

## 4. Cellular Functions of STK16 

There are a few studies indicating that STK16 binds different interactors (GlcNAck/DRG1/MAL2/WDR1/Actin) and participates in various physiological activities, including the TGF-β signaling pathway and TGN protein secretion and sorting, as well as cell cycle and Golgi assembly regulation.

### 4.1. STK16 Participates in the TGF- β Signaling Pathway

TGF-β signaling pathways include both Smads-dependent and Smads-independent pathways. Due to the diversity of TGF-βs, TGF-β family receptors, Smads, and DNA-elements, as well as the different interaction patterns between various proteins [69,70,71,72], TGF-β signaling pathways can induce very diverse physiological and pathological responses [73,74,75,76], including immunity [77], cancer [78], fibrosis [79], and hematopoietic homeostasis [80]. 

It was reported that STK16 possesses DNA-binding ability independent of its kinase activity. It binds to GC-rich elements of TGF-β responsive CNP (C-type natriuretic peptide) promoter and VEGF (vascular endothelial growth factor) promoter to activate them [15]. Moreover, as well as the mRNA and protein levels of STK16 responding to TGF-β treatment, elevated STK16 could also increase the mRNA level of STK16, which indicates that there may be a positive feedback loop between the transcriptional activity of STK16 and its own protein level [19]. 

It was reported that STK16 is able to enter the nucleus. After Golgi disorganization by treatment with BFA or nocodazole, STK16 lost the Golgi localization and translocated into the nucleus in NIH3T3 cells in a kinase activity independent way [19]. In HT1080 cells, the kinase-dead mutant STK16-E202A can also induce the expression of VEGF, which is consistent with the fact that STK16 activates VEGF gene promoter transcriptional activity independently of its kinase activity [15]. Moreover, higher STK16 transcriptional activity was induced by BFA or nocodazole treatment in STK16 wild type and E202A overexpressed cells, which is consistent with previous research that STK16 nuclear translocation could up-regulate STK16 transcription [15]. In fact, the alternative splicing of STK16 is related to the differential transcription activity of ELK1 [81]. The change in the exon inclusion ratio of STK16 was related to ELK1 transcription, which mainly depends on the fourth exon located in the kinase domain.

In conclusion, STK16 may be a nuclear factor that enters the nucleus following certain stimuli, participating in the TGF-β signaling pathway and regulating transcriptional activity. However, very low nuclear localization of endogenous STK16 has been detected in unstimulated cells, which may be caused by the formation of complexes between STK16 and other proteins. In addition, the nuclear localization between STK16 in various species are also likely different. Moreover, most current commercial STK16 antibodies have limited specificity or application, and development of more specific STK16 antibodies is crucial for further investigations of STK16.

### 4.2. STK16 and GlcNAcK (N-Acetylglucosamine Kinase)

GlcNAc (N-Acetylglucosamine) is an important component of bacterial cell wall peptidoglycan [82], fungal cell wall chitin [83], and the extracellular matrix of animal cells [84]. In animal cells, GlcNAc also regulates the distribution of glycoproteins at the cell surface through glycosylation and further regulates cell signaling transduction [85,86]. Mammalian GlcNAcK (N-Acetylglucosamine Kinase) converts GlcNAc from lysosomal degradations or nutritional sources to GlcNAc-6-phosphate. Eventually, GlcNAc-6-phosphate can enter the catabolic pathway to form fructose 6-phosphate [87] or enter the anabolic pathway to synthesize UDP-GlcNAc [88]. Moreover, UDP-GlcNAc serves as a donor for various glycoconjugates and glycans to provide GlcNAc [89,90]. The metabolic imbalance of GlcNAc causes serious damage to cells [91,92]. In addition, GlcNAcK also has some non-traditional functions. It is highly expressed in neurons [93], and the interaction of GlcNAcK-Dynein-Golgi could regulate the growth of axons [94] and dendrites [95] of neurons. Moreover, the interaction of GlcNAcK-Dynein-Lis1-NudE1 plays an important role in the prophase and metaphase of mitosis [96]. Meanwhile, GlcNAcK also has different subnuclear distribution [97]. 

GlcNAcK was reported as another STK16 interactor [20], although GlcNAcK is not a substrate of STK16, and STK16 does not affect GlcNAcK activity in vivo or in vitro. Although GlcNAcK is unable to regulate STK16 autophosphorylation, it greatly affects the phosphorylation effect of STK16 on its substrates. In the interphase NIH-3T3 cells, endogenous GlcNAcK is located predominantly in the perinuclear area and the cell periphery. Endogenous STK16 is associated mainly with the Golgi, co-localizing partially with GlcNAcK. However, when GlcNAcK was overexpressed with STK16, they highly colocalized in a vesicular pattern. 

The STK16/GlcNAcK interaction pattern unravels their new function of translocation. GlcNAcK regulates the Golgi complex transport by the dynein motor [95]. Therefore, as an interactor of the dynein complex, GlcNAcK may be involved in the transportation of STK16. In particular, STK16 is a Golgi resident protein [11,19,20,21]. The interaction between STK16 and GlcNAcK causes the redistribution of GlcNAcK and may alter the metabolic balance of GlcNAc, thus participating in glucose metabolism. It is also possible that STK16 plays a role in the non-traditional functions of GlcNAcK, which needs further investigation.

### 4.3. STK16 and MAL2/WDR1 (Myelin and Lymphocyte Protein 2/ WD Repeat Containing Protein-1)

MAL2, a raft protein of the MAL proteolipid family, contains four transmembrane domains and regulates the transcytotic delivery pathways of various proteins [98]. It is necessary for basolateral-to-apical transcytosis in HepG2 cells, delivering membrane-associated proteins and exogenous cargo, such as pIgA (polymeric immunoglobulin A-receptor) and glycosylphosphatidylinositol-anchored protein CD59 [99,100,101]. The interaction between STK16 and MAL2, discovered by split-ubiquitin yeast two-hybrid assays, regulates secretory soluble cargo sorting into the constitutive secretory pathway at the TGN (trans-Golgi network) in hepatocytes [102]. Overexpression of STK16 kinase-dead mutant E202A or knockdown of MAL2 in polarized hepatic WIF-B cells impairs soluble cargo secretion, resulting in decreased secretion of albumin and haptoglobin. The synthesis and process of albumin in E202A-overexpressing cells or MAL2-knockdown cells are not interfered, but they are not secreted as in wild-type cells. Instead, they are directly degraded in lysosomes after being delivered from ER to the Golgi. Therefore, albumin detection was recovered in E202A-overexpressing cells or MAL2-knockdown cells after lysosomal deacidification. Thus, STK16 and MAL2 may play a key role in protein sorting at the TGN. 

Consistent with the interaction between STK16 and MAL2, several proteins interact with MAL2 to influence secretion. For example, Formin INF2, associating with cdc42 and MAL2, regulates basolateral-to-apical transcytosis and lateral lumen formation in HepG2 cells [103]. In addition, MAL2 controls vesicle transportation through interaction with TPD52 (tumor protein 52)-like proteins [98]. In Candida albicans, STK16 homologue named CaENV7 interacts with two TGN-resident proteins Imh1p and Arl1p. Imh1 is phosphorylated by CaEnv7 and their interaction affects the localization of Imh1, thus maintaining the morphology and function of TGN and having an influence on protein secretion and delivery [104]. Moreover, the overexpression of STK16 leads to abnormal endbud formation in the mammary gland during puberty, which may be caused by abnormal cytokine secretion, and STK16 may regulate the process [18]. In a word, STK16, highly expressed in the liver with constitutive kinase activity, regulates the secretory function through the interaction with MAL2. 

Furthermore, increased E202A levels further impaired albumin secretion, suggesting that STK16 kinase activity is required for soluble secretion [102]. Recently, WDR1 was reported as a possible substrate of STK16, regulating constitutive secretion [105]. WDR1, referred to as AIP1 (actin-interacting protein 1), was first identified as an actin-binding protein. As the binding site of actin and cofilin, its conserved WD40 repeat sequences regulate the morphology and function of the cytoskeleton [106,107,108], and the hepatic secretion requires actin remodeling [109,110].

Together with MAL2 and WDR1, it is likely that STK16 regulates the constitutive secretion in the hepatocyte. Although STK16 interacts with MAL2 and is a candidate kinase for WDR1, further studies are necessary to confirm whether MAL2 is the substrate of STK16 and whether WDR1 is the direct substrate. 

### 4.4. STK16 and DRG1 (Developmentally Regulated GTP Binding Protein 1)

DRG1 (developmentally regulated GTP-binding protein 1) belongs to the DRG family that contains DRG1 and DRG2 [111], which is significantly expressed in the embryonic brain and downregulated during development [112,113]. DRG1 is highly expressed in various embryonic tissues but is greatly decreased in newborn animals [112]. In addition, in mouse and human, DRG1 specifically binds to SCL/TAL (stem cell leukemia/T-cell acute lymphoblastic leukemia (T-ALL) 1), a highly conserved basic helix-loop-helix transcription factor participating in cell growth and differentiation and required for normal hematopoietic development [114,115]. 

Evidence from pull-down experiments has shown the specific interaction between DRG1 and STK16 [17]. Furthermore, the 65 amino acids at the N-terminus of DRG1 are sufficient for STK16 binding, and DRG1 is phosphorylated at Thr100 within the GTPase domain by STK16. In contrast, DRG2, which has a Cys100 instead of Thr, does not interact with STK16 [17]. Similarly, although DRG1 and DRG2 are both widely expressed in human and mouse tissues and their sequences share 62% identity [111], they function differently by binding distinct regulatory proteins. For instance, DFRPs (DRG family of regulatory proteins) maintain the stability of DRGs through specific binding, for example, DFRP1 binds DRG1, and DFRP2 binds DRG2, respectively [111]. In other species, the transcription or stability of DRG1 and DRG2 transcripts were also found to be regulated differently [116]. Moreover, DRG1 was identified as a direct microtubule binding protein, which drives microtubules into bundles and stabilizes microtubules. In DRG1-knockdown cells, the progression from prophase to anaphase is delayed and spindle formation is slowed down [117], whereas DRG2 interacts with tau as a mechanism for regulating microtubule dynamics in HeLa cells [118].

These observations indicate that DRG1 is a specific interactor and substrate of STK16. However, further experiments are needed to reveal the mechanism by which STK16 regulates DRG1 signaling and its related cellular functions.

### 4.5. STK16 and Actin

Previous studies showed that STK16 can interact with actin to regulate the Golgi apparatus dynamics. Although the interaction between STK16 and actin is independent of its kinase activity, STK16 regulates actin dynamics in a concentration and kinase activity-dependent way (Figure 3) [21]. 

As an important membranous organelle of eukaryotes, Golgi apparatus is a central component of the endomembrane system and plays key roles in trafficking, post-translational modifications of proteins, and lipid biosynthesis [119]. It has been widely reported that actin and actin-binding proteins that localize on the Golgi apparatus could regulate the structure and function of the Golgi [120,121]. For example, GRASP65 (Golgi reassembly and stacking proteins 65), distributed on the peripheral membrane of the Golgi apparatus, interacts with the actin-binding protein Mena and leads to its own homologous oligomerization, which mediates the stacking of the Golgi cisternae and ribbon formation [122,123,124]. Furthermore, GRASP55 and GRASP65 are phosphorylated to be depolymerized and control Golgi fragmentation in mitosis [125,126]. During the highly dynamic and precisely regulated mitosis, the Golgi structure experiences fragmentation and remodeling, which is critical for mitotic progression [127]. Therefore, STK16 knockdown or kinase inhibition could arrest cells in prometaphase and cytokinesis and delay mitotic progressions [21,40]. 

Besides actin and actin-binding proteins, the assembly, morphology, localization, and orientation of the Golgi apparatus are also correlated to microtubules and microtubule-related proteins [128,129,130], and the regulation between cytoskeleton with the Golgi complex has been a hot topic for mitotic regulation studies [131]. STK16 interacts with cytoskeleton to regulate the Golgi structure, whereas the detailed regulatory mechanisms are unexplored, such as the binding mechanism of STK16 and actin, which step of Golgi apparatus assembly STK16 and actin are involved in, and how STK16 and actin regulate the Golgi assembly. The elucidation of these questions will contribute to our understanding of the regulation between the cytoskeleton and the Golgi apparatus.

## 5. Conclusions and Perspectives

As a membrane-associated protein with constitutive kinase activity, STK16 has been found to participate in various signaling pathways and cellular process regulations in eukaryotes, such as TGF-β signaling pathway, TGN protein secretion and sorting, as well as the Golgi assembly and cell cycle (Figure 4). Moreover, we also know that STK16 undergoes at least two post-translational modifications, fatty acylation, and phosphorylation. However, what serves as STK16 kinase activity switch, what the upstream regulators and kinases are, and what the specific localization of different forms of phosphorylated STK16 during mitosis are all unclear so far, which encourages people to conduct further mechanistic studies to unravel its cellular functions and regulation mechanisms so that we can have a more comprehensive understanding about this kinase.

## Figures and Tables

**Figure 1 ijms-20-01760-f001:**
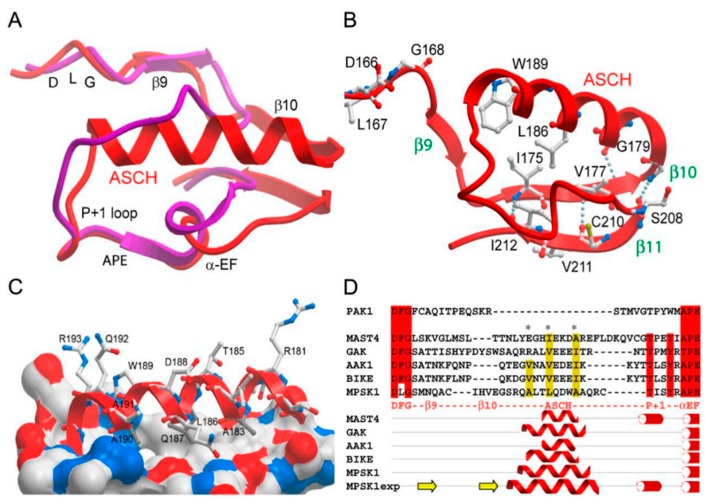
STK16 activation loop architecture [17]. (**A**) Structural overlay of the activation loop of active Aurora A (PDB ID code: 1OL7) (shown in magenta) with MPSK1 (shown in red). The main structural elements are labeled. DLG motif (Asp-Leu-Gly) is the initiation of the activation segment of STK16. APE motif (Ala-Pro-Glu) is the end of the activation segment of STK16 and P + 1 loop is before of the APE motif. (**B**) Hydrogen bonds and hydrophobic interactions stabilizing the activation segment of MPSK1 are shown as dotted lines and the residues involved in stabilization are shown in ball-and-stick representation. The interacting β sheet (β11), the P + 1 loop, and the ASCH, as well as the helix αEF, are labeled. (**C**) Interface of the ASCH interacting with the lower kinase lobe. Hydrophobic residues are indicated as solid white surfaces. (**D**) Prediction of similar activation loop helices present in the kinome. Secondary structure elements predicted to be smaller than three residues have been deleted. The experimentally determined secondary structure (MPSK1exp) and the predicted one of MPSK1 are also shown. The activation segment helix and helix αEF were predicted accurately, whereas the β sheet secondary structure was not recognized by the prediction program. One representative member of the MAST kinase family predicted to contain an activation loop helix is also shown. Hydrophobic residues, buried in the interface between the ASCH and the lower kinase lobe, are indicated (^∗^) in the sequence alignment. The ninth and tenth β sheets are indicated by yellow arrows. ASCH is indicated by red wavy line. P + 1 loop and helix αEF are indicated by red bar with white circle. Figure was adapted from ref [17].

**Figure 2 ijms-20-01760-f002:**
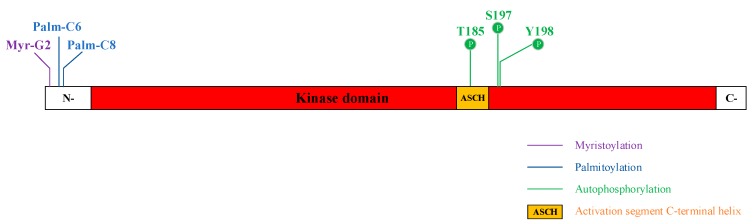
Illustration of posttranslational modification of STK16.

**Figure 3 ijms-20-01760-f003:**
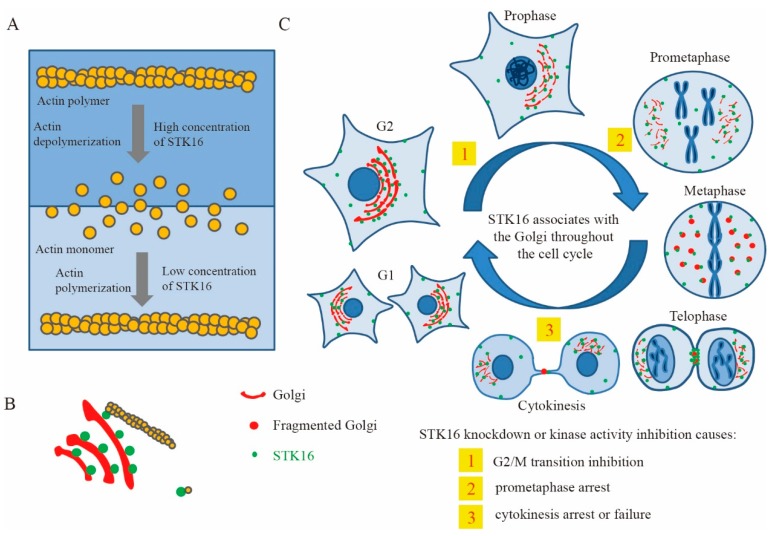
The model of STK16 functions in actin, the Golgi, and cell cycle regulation [21]. (**A**) STK16 kinase is a novel actin binding protein that regulates actin polymerization and depolymerization in a concentration-dependent manner. (**B**) STK16 localizes to the Golgi and bridges the Golgi with actin. (**C**) STK16 plays important roles in G2/M transition and mitotic progression as well as cytokinesis. Figure was adapted from [21].

**Figure 4 ijms-20-01760-f004:**
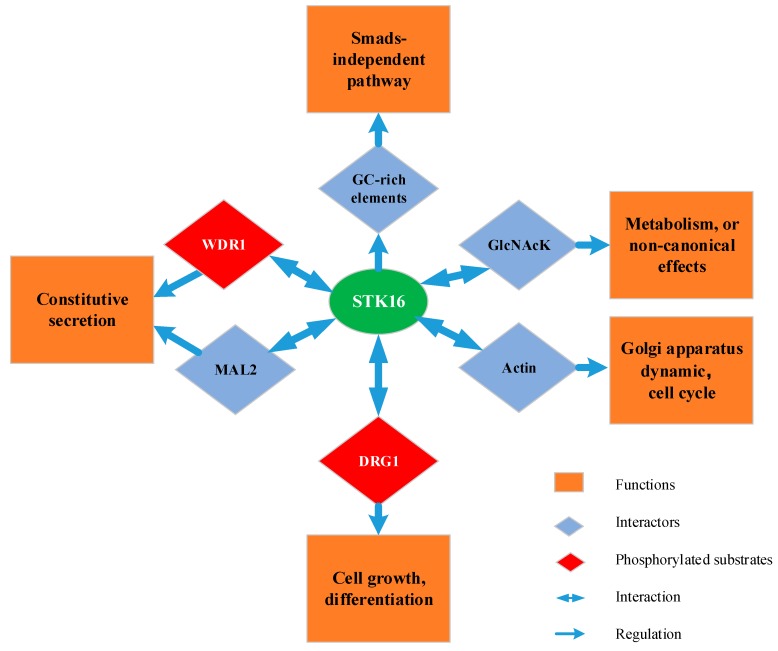
STK16 pathways.

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
