# Peer review of "Serine/Threonine Protein Kinase STK16"

_ijms, 2019, doi:10.3390/ijms20071760_

Round 1

Reviewer 1 Report

This review by J. Wang, et al. summarizes information currently available on the structure and cellular pathways of Ser/Thr protein kinase STK16. Overall, the manuscript is well structured and relatively coherent. However, there are some minor inconsistencies and misworded sentences that need to be improved to make the text comprehensible for the future readers:

1.      The language needs editing throughout the manuscript – just to list some of the examples: in abstract, “researches” should be replaced with “studies”; in the inset of Figure 1, “rule” should be replaced with “ruler”. In lane 138, statement “lysine 49 is the predicted ATP‐binding site” should be corrected: one residue cannot serve as the site for ATP binding, the authors probably meant that Lys49 is a catalytically important/conserved residue residing within the ATP-binding site of STK16. The sentence in lanes 236-238 (“The interaction between STK16 and GlcNAcK causes the redistribution of GlcNAcK and may alter its metabolic balance, thus participates in glucose metabolism”) is not understandable. The beginning of sentence in lanes 298-300 “These indicate that” should be changed to “These observations indicate that” or similar.

2.      In section 3.2, the authors describe the plausible mechanism of activation of STK16 and compare the latter to the well-known PKs. However, the comparison to Akt is partially incorrect: the authors state that “Akt is a serine/threonine kinase that is activated by insulin and other growth factors”. This is true in the context of cellular signalling upstream of Akt activation, yet not true mechanistically: neither insulin nor growth factors bind to Akt directly. Instead, Akt is activated by phosphorylation on multiple residues, of which one is carried out by PDK1. Also, the sentence at the end of section “Since Serine/Threonine kinases usually possess multiple phosphorylation residues on the activation loop that have distinct effects on autophosphorylation” requires a more substantial reference than ref 59 currently provided, as the latter focusses on the single example of Stk1 from S. aureus.

3.      The sentence at the beginning of section 4.1 is confusing (“Due to the diversity of TGF‐βs, receptors, Smads, and DNA‐elements, as well as the different interaction patterns between various proteins, TGF‐β signaling pathways can induce very diverse physiological responses [64‐67], including immunity [68], cancer [69], fibrosis [70] and hematopoietic homeostasis [71].”). First, the word “receptors” needs to be specified – there might be indeed myriads of different receptors in a cell, yet not all of these are related to TGF_beta signalling. Second, the list of “physiological responses” seems somewhat inadequate: cancer cannot be really considered as a physiological response, it is a major disease.

4.      The statement that “most current commercial STK16 antibodies have limited specificity or application” (lanes 209-211) needs a proper reference.

5.      Figure 2 should be enlarged, or the font size of the text presented in the figure should be increased. Also, as this illustration originates from the previous publication by the same authors (https://www.nature.com/articles/srep44607), the reference to the previous publication should be provided.

6.      The value of the manuscript could be improved by introducing some extra information on STK16. For instance, the positioning of the NAK family of protein kinases in the context of human kinome tree (http://kinase.com/wiki/index.php/Kinase_classification), the substrate specificity (e.g., preferred recognition sequences of substrates flanking phosphorylatable residue) and available inhibitors of STK16 could be mentioned. In addition, as the authors discuss briefly the molecular architecture of this kinase, incorporation of a figure showing crystal structure of STK16 and highlighting the discussed features (such as the atypical activated loop or residues undergoing post-translational modifications) would be helpful. I would also recommend including list of abbreviations into the structure of manuscript.

7.      Within the list of references, several publications are listed originating from over 10 years ago. Surely, more recent references would be more appealing for the readers. Furthermore, there are issues with formatting of some references: the names or the initials of some authors seem to be missing (e.g., refs 97 or 100), and full names of several authors are missing in case of ref 118 (https://pubs.acs.org/doi/10.1021/acschembio.6b00250).

Author Response

We thank both reviewers for their comments and suggestions, which are very helpful for us to improve our manuscript. We have now addressed them to the fullest extent possible. We have used track change or red font to highlight the major changes in the revised manuscript. We didn’t track change some recurring small changes for better readability, such as amino acid abbreviations. Below are the point-to-point responses.

This review by J. Wang, et al. summarizes information currently available on the structure and cellular pathways of Ser/Thr protein kinase STK16. Overall, the manuscript is well structured and relatively coherent. However, there are some minor inconsistencies and misworded sentences that need to be improved to make the text comprehensible for the future readers:

1.      The language needs editing throughout the manuscript – just to list some of the examples: in abstract, “researches” should be replaced with “studies”; in the inset of Figure 1, “rule” should be replaced with “ruler”. In lane 138, statement “lysine 49 is the predicted ATPbinding site” should be corrected: one residue cannot serve as the site for ATP binding, the authors probably meant that Lys49 is a catalytically important/conserved residue residing within the ATP-binding site of STK16. The sentence in lanes 236-238 (“The interaction between STK16 and GlcNAcK causes the redistribution of GlcNAcK and may alter its metabolic balance, thus participates in glucose metabolism”) is not understandable. The beginning of sentence in lanes 298-300 “These indicate that” should be changed to “These observations indicate that” or similar.

---------Thanks for pointing out all these mistakes. We have corrected all of them in the revised manuscript. Because we have added a new figure 1, the line numbers have changed in the revised manuscript. The previous “lane 138” has been changed to lane 169. The previous “lanes 236-238” has been changed to lanes 265-267, the previous “lane 298-300” has been changed to lanes 334-336.

2.      In section 3.2, the authors describe the plausible mechanism of activation of STK16 and compare the latter to the well-known PKs. However, the comparison to Akt is partially incorrect: the authors state that “Akt is a serine/threonine kinase that is activated by insulin and other growth factors”. This is true in the context of cellular signalling upstream of Akt activation, yet not true mechanistically: neither insulin nor growth factors bind to Akt directly. Instead, Akt is activated by phosphorylation on multiple residues, of which one is carried out by PDK1. Also, the sentence at the end of section “Since Serine/Threonine kinases usually possess multiple phosphorylation residues on the activation loop that have distinct effects on autophosphorylation” requires a more substantial reference than ref 59 currently provided, as the latter focusses on the single example of Stk1 from S. aureus.

---------Yes, we agree with the reviewer that the statement was incorrect. We have now deleted this sentence because it is not necessary here. Thanks for pointing it out. Moreover, we added NIK as an additional example to illustrate the physiological importance of constitutively active kinases. The previous “lane 150-153” has been changed to lanes 182-189. We have added reference 62 and 63 as other examples.

3.      The sentence at the beginning of section 4.1 is confusing (“Due to the diversity of TGFβs, receptors, Smads, and DNAelements, as well as the different interaction patterns between various proteins, TGFβ signaling pathways can induce very diverse physiological responses [6467], including immunity [68], cancer [69], fibrosis [70] and hematopoietic homeostasis [71].”). First, the word “receptors” needs to be specified – there might be indeed myriads of different receptors in a cell, yet not all of these are related to TGF_beta signalling. Second, the list of “physiological responses” seems somewhat inadequate: cancer cannot be really considered as a physiological response, it is a major disease.

--------------Sorry that our previous statements are not accurate. We have now corrected both of them. The old “lane 183-187” has been changed to lanes 219-223.

4.      The statement that “most current commercial STK16 antibodies have limited specificity or application” (lanes 209-211) needs a proper reference.

--------------We are sorry but there is actually no published reference for this. The STK16-related studies are very limited and only a few of them have used antibodies. The rabbit polyclonal antibody used in Katsuki Kurioka and Bárbara Guinea’s studies were custom-made. We have bought a few commercially available antibodies, including #SAB1406692(Sigma-Aldrich) and ab37975(Abcam), and found that most of them work for purified STK16 proteins, but not endogenous STK16. Below are some examples.

5.      Figure 2 should be enlarged, or the font size of the text presented in the figure should be increased. Also, as this illustration originates from the previous publication by the same authors (https://www.nature.com/articles/srep44607), the reference to the previous publication should be provided.

--------------------- Thanks for this suggestion. We have modified the figure to make it clear and inserted the corresponding reference. The figure number has been changed to figure 3 because we have inserted a new figure 1.

6.      The value of the manuscript could be improved by introducing some extra information on STK16. For instance, the positioning of the NAK family of protein kinases in the context of human kinome tree (http://kinase.com/wiki/index.php/Kinase_classification), the substrate specificity (e.g., preferred recognition sequences of substrates flanking phosphorylatable residue) and available inhibitors of STK16 could be mentioned. In addition, as the authors discuss briefly the molecular architecture of this kinase, incorporation of a figure showing crystal structure of STK16 and highlighting the discussed features (such as the atypical activated loop or residues undergoing post-translational modifications) would be helpful. I would also recommend including list of abbreviations into the structure of manuscript.

--------------------- This is a really good suggestion. We have added additional information to the text (lines 72-73, lines 163-165, and lines 121-125 in the revised manuscript) and also added a new figure to illustrate the STK16 structure (New figure 1). We totally agree with the reviewer that this information will be very helpful to improve the readability of our manuscript. Thanks a lot for all these helpful suggestions.

7.      Within the list of references, several publications are listed originating from over 10 years ago. Surely, more recent references would be more appealing for the readers. Furthermore, there are issues with formatting of some references: the names or the initials of some authors seem to be missing (e.g., refs 97 or 100), and full names of several authors are missing in case of ref 118 (https://pubs.acs.org/doi/10.1021/acschembio.6b00250).

---------------------We have replaced some of the old references with newer ones. However, since there are very limited studies about STK16, we kept all STK16-related old references. For the formatting problem, we have corrected them. Thanks a lot for pointing them out!

Reviewer 2 Report

The manuscript presents a lesser known, but interesting protein kinase. The article is well written, but has some editing problems (see row 61). Pay attention to add proper space between words. The fonts are different across the text and the formatting is not always the proper one according to the journal style. Cysteine, threonine and other aminoacids could be easily abbreviated.

I think the manuscript could discuss more the pharmacological potential of this PK and its inhibitors. The section 89-91 can also be developed more.

I recommend the authors to target their manuscript not only for the people with a very good background in cell signaling, but also for scientists with various other backgrounds. Some minor modification could really improve the paper.

Author Response

We thank both reviewers for their comments and suggestions, which are very helpful for us to improve our manuscript. We have now addressed them to the fullest extent possible. We have used track change or red font to highlight the major changes in the revised manuscript. We didn’t track change some recurring small changes for better readability, such as amino acid abbreviations. Below are the point-to-point responses.

Reviewer#2

The manuscript presents a lesser known, but interesting protein kinase. The article is well written, but has some editing problems (see row 61). Pay attention to add proper space between words. The fonts are different across the text and the formatting is not always the proper one according to the journal style. Cysteine, threonine and other amino acids could be easily abbreviated.

------------Thanks for the reviewer’s positive comments and helpful suggestions. We have carefully edited the whole manuscript and corrected these mistakes. We didn’t track change these changes for better readability.

I think the manuscript could discuss more the pharmacological potential of this PK and its inhibitors. The section 89-91 can also be developed more.

------------This is a really good suggestion. We have also expended section 89-91 and added additional information to the text (page 3 and lines 109-125 in the revised manuscript).

I recommend the authors to target their manuscript not only for the people with a very good background in cell signaling, but also for scientists with various other backgrounds. Some minor modification could really improve the paper.

-------------Yes, we should have written it in a way that is easily understandable for readers with broader background.